

# Light-absorbing snow impurities: Nine years (2016-2024) of snowpack sampling close to Sonnblick Observatory, Austrian Alps

Daniela Kau[1], Marion Greilinger[2], Andjela Vukićević[1], Jakub Bielecki[1], Laura Kronlachner[1], Anne Kasper-Giebl[1]

[1]Institute of Chemical Technologies and Analytics, TU Wien, Vienna, 1060, Austria
[2]formerly associated with Section Climate Monitoring and Cryosphere, GeoSphere Austria, Vienna, 1190, Austria

*Correspondence to*: Daniela Kau (daniela.kau@tuwien.ac.at)

**Abstract.** We present chemical analysis data of the seasonal snow cover focusing on the light-absorbing snow impurities elemental carbon and mineral dust collected at a high-alpine glacier field close to Sonnblick Observatory. Sampling covered the whole winter accumulation periods between 2016 and 2024. The co-occurrence of mineral dust leads to an underestimation of elemental carbon quantified via thermal-optical analysis. To minimise the bias, we apply a linear laser correction, leading to a median increase in elemental carbon by 63 % for single samples and up to 8.3 % for entire snowpacks collected at the end of the accumulation period. Average concentrations for elemental carbon and water-insoluble organic carbon were 11.1±2.5 and 458±215 ng g$^{-1}$, respectively. Using the interference introduced by mineral dust, we identify mineral dust layers and find very good agreement with a complementary method based on calcium concentrations and the pH. Based on thermal-optical analysis and an average share of iron in mineral dust mass of 4 %, the approximated mineral dust input ranged up to 2100 mg m$^{-2}$. Results agree well with gravimetric results.

## 1 Introduction

The seasonal snow cover acts as a reservoir for deposited compounds. Sampling and analysing this repository allow to assess the input of various species during the accumulation period. Compounds of interest include light-absorbing impurities (LAI) in snow, also known as light-absorbing snow impurities (LASI) or light-absorbing particles (LAPs), due to their direct and indirect radiative forcing. After deposition on snow surfaces, LASI affect the hydrological cycle via reduction of the albedo and earlier melt of the snow cover and can impact snowpack stability (e.g., Warren and Wiscombe 1980; Tuzet et al., 2020; Réveillet et al, 2022; Dick et al., 2023). Upon melt, insoluble light-absorbing particles are enriched on the snow surface, amplifying their adverse effects (Doherty et al., 2013). Mineral dust and elemental carbon pose important and widely studied representatives of LASI, especially in background regions like the Alps or the Arctic (e.g., Roussel et al., 2025; Di Mauro et al., 2019; Doherty et al., 2010; Svensson et al., 2018; Thevenon et al., 2009) or at the glaciers in the Himalayas and the Tibetan Plateau, which are close to source regions and thus show even higher input (e.g., Gul et al., 2022; Li et al., 2018; Thind et al., 2019).





Global mineral dust emission is dominated by North African source regions, has increased substantially since pre-industrial times and was estimated to be 29±8 Tg between 1981 and 2000 (Kok et al., 2021; Kok et al., 2023). After long-range transport, mineral dust is deposited regularly in the Alps and may reach up to northern Europe or the Arctic (e.g., Meinander et al., 2023; Greilinger and Kasper-Giebl, 2021; Barkan and Alpert, 2010). Deposited on snow, it increases alkalinity and affects the biogeochemical cycle by introducing several elements, e.g., Ca, Fe and P (e.g., Lafon et al., 2006; Nielson and Brahney, 2025;

Rogora et al., 2004). Consequently, the pH and $Ca^{2+}$ concentration (Greilinger et al., 2016) or the elemental composition including Fe, Ca, Al, Si and Ti can be used to determine mineral dust (e.g., Li et al., 2017; Telloli et al., 2018). The possibility to assess the Fe loading on filters influenced by mineral dust from thermal-optical analysis (Kau et al., 2022) offers a starting point to estimate mineral dust from thermal-optical analysis data.

Elemental carbon is emitted by incomplete combustion and is estimated to be the second strongest climate forcer from human

emission ranking only behind $CO_2$ (Bond et al., 2013). Various analytical techniques can be applied for the quantification of elemental carbon (or black carbon, see Petzold et al., 2013 for differentiation), including laser-induced incandescence, optical methods or thermal-optical analysis (Schwarz et al., 2012; Grenfell et al., 2011; Birch and Cary, 1996). The latter is the reference method for the quantification of elemental carbon in ambient air samples (DIN e.V., 2017) and is commonly applied to filters loaded with insoluble particles from snow samples. The co-occurrence of mineral dust leads to an interference in the

quantification of elemental carbon and adapted evaluation of measurement data was proposed (Wang et al., 2012; Gul et al., 2018). A comprehensive discussion of the impact of this bias on elemental carbon results in seasonal snow covers is still lacking.

We present a multi-year time series of LASI (elemental carbon and mineral dust) monitoring at a glacier field close to Sonnblick Observatory in the Austrian Alps. Sampling started in 2016, and elemental carbon and water-insoluble organic

carbon are quantified via thermal-optical analysis. Care is taken to use a suitable laser correction method to account for a possible bias introduced by mineral dust. We discuss the impact of mineral dust on elemental carbon concentrations for single samples and entire snowpacks. Based on thermal-optical analysis, we introduce a method to identify mineral dust layers and to approximate the mineral dust concentration and deposition. To do so, a conversion factor based on the Fe content of mineral dust samples is deduced.

## 2 Sampling, sample preparation and analysis

### 2.1 Snow sampling

Snowpacks were collected at the glacier field Goldbergkees (GOK) at an elevation above 3000 m a.s.l. in the National Park Hohe Tauern, located in the Austrian Alps. The high-alpine site is distant to anthropogenic activities and in the vicinity of the Global Atmosphere Watch (GAW) station Sonnblick Observatory. The sampling procedure is described in detail in Greilinger

et al. (2016). An overview is given here: At the end of the winter accumulation period (end of April or early May), snowpacks were sampled in increments of 20 cm for carbonaceous compounds (referred to as TOA-profile) and 10 cm for $Ca^{2+}$



concentration and pH (referred to as IC-profile). The samples were cut using stainless-steel cylinders with a diameter of 5.6 and 5.0 cm, respectively. The larger increments for the TOA-profile were a compromise between depth resolution and sufficient analyte loadings. At the same time this allowed to use samples collected for the determination of snow density. The

samples were transported to the laboratory frozen in polyethylene bags and kept frozen until processing. The profiles were assigned to the year of sampling (2016-2024), i.e., the profile of 2016 corresponds to the winter accumulation period 2015/2016.

In 2023, sampling had to be postponed due to challenging weather conditions, leading to a rather late sampling date. Melting of snow on the wall of the snow pit was observed and wash down of compounds from upper to lower layers cannot be excluded.

Challenging weather conditions stopped the sampling in 2019, when only 3.2 of 4.4 m of the snow cover could be sampled. In the residual years, the entire snow cover could be sampled.

## 2.2 TOA-profile

### 2.2.1 Sample preparation

Sample preparation is described in detail by Meinander et al. (2022) within the sections referring to TU Wien. In short, after

melting the snow sample of the TOA-profile in a glass beaker using a microwave, it was filtrated onto a quartz fibre filter (PALLFLEX® Tissuquartz™, Pall Laboratory), leading to a loaded area with a diameter of 16 mm. After careful evaluation of literature and own experiments we decided not to use a coagulant to increase filtration efficiency. A detailed discussion can be found in Appendix A. After filtration, few samples (n=10, ~6%) showed low particulate filter loadings. To reach loadings above the limit of detection, depth resolution was sacrificed, and the subsequent sample was loaded on the same filter. Contrary,

one dust-laden sample of 2016 included high particulate loading and was filtrated on two separate filters to prevent loss of particles during handling and analysis. For all samples from 2016, a punch with a diameter of 15 mm was used and the result calculated for the entire loaded area. For some of these samples, a darkening around the rim of the loaded area was observed and underestimation of the results is possible. To ensure quantification of carbon in the entire sample, from 2017 on a punch with a diameter of 17 mm was used.

All samples were cut in half to fit into the instrument, both parts were analysed and considered. Masses of water-insoluble particles on the filters were determined gravimetrically for the years 2017-2022 and 2024.

### 2.2.2 Thermal-optical analysis

Water insoluble total carbon (WinsTC), split into water insoluble organic carbon (WinsOC) and elemental carbon (EC), was quantified via thermal-optical analysis (TOA) using a Lab OC-EC Aerosol Analyzer (Sunset Laboratory Inc.). Samples were

measured with the EUSAAR2 protocol (Cavalli et al., 2010) in the program OCEC834 and evaluated with the program Calc453 (both Sunset Laboratory Inc.). For blank correction, several method blanks were prepared each time samples were processed. Snowpack layers correspond to water equivalents of 70-290 g, consequently between 100 and 200 g of ultrapure water were



drawn through a preheated quartz fibre filter, using the identical procedure and labware as for the samples. These blanks showed only signals for WinsOC; hence only WinsOC and WinsTC loadings were blank corrected with the method blanks of

the respective year (between 1.99 and 3.53 µg/cm²). Limits of detection were calculated as the threefold standard deviation of the blanks and ranged between 0.98 and 3.76 µg/cm² for both WinsOC and WinsTC for the different years. The limit of detection for EC was determined as the threefold standard deviation of repeated analysis of a lightly loaded $PM_{10}$ filter collected at Sonnblick Observatory (n=10). The result was 0.20 µg/cm² and agrees with the lower limit of the measurement range of the instrument given by the manufacturer. For 2016 only WinsTC is available due to insufficient quality of the laser signal. Quality

assurance included the daily analysis of a sucrose standard and participation in international comparisons conducted annually within ACTRIS (Aerosol, Clouds and Trace Gases Research Infrastructure) since 2021.

### 2.2.3 Inductively coupled plasma-optical emission spectrometry

Quantification of Fe via inductively coupled plasma-optical emission spectroscopy (ICP-OES) was done using a 5110 ICP-OES instrument (Agilent). Selected dust-laden filters from the TOA-profiles were subjected to microwave-assisted acid

digestion prior to analysis, as described in Kau et al. (2022). Mineral dust from long-range transport obtained from snow and rime and local dust samples recovered from bare rock were also analysed this way, digesting about 12 mg. For duplicates and triplicates of these dust samples, the entire digestion process (weighing in and digesting) was repeated. More information on these samples can be found in Appendix B.

Measurement conditions included a viewing height of 10 mm, a read time of 10 s, an RF power of 1.20 kW, and nebulizer,

plasma and auxiliary gas flows of 0.70, 12.0, and 1.00 L min$^{-1}$, respectively. Fe was quantified at 238.204 nm. External standards were prepared using ICP multi-element standard solution VIII (Certipur®, Supelco®, Sigma-Aldrich). Limit of detection corresponded to 0.1 µgFe/cm². Eu (plasma standard solution, Specpure®, Thermo Scientific) was used as internal standard and added to all standards and samples in a final concentration of 10 mg/l. For quality assurance, quintuplicate digestion and analysis of NIST® Standard Reference Material® 2709 San Joaquin Soil was conducted. Average recovery for

Fe was 97% with a standard deviation of 7% and a median of 99%.

### 2.3 IC-profile

For the quantification of $Ca^{2+}$ using ion chromatography (IC; Dionex Aquion, Thermo Fisher) and the electrochemical determination of the pH value (InLab® Pure Pro--ISM, Mettler Toledo), samples of the IC-profile were melted inside the polyethylene bags at room temperature and filled into vials. Samples with high particulate loadings were filtrated using a

syringe filter (PALL ACRODISC® 0.2 µm Supor®) prior to analysis. Instrumentation for $Ca^{2+}$ quantification included Dionex Ion Pac CS16 and CG16 columns and 30 mM methanesulfonic acid as eluent. Measurements and evaluations were done in Chromeleon 7 software (Thermo Fisher Scientific). Quantification in IC was done using external standards (mixture and dilution of 1000 ppm Ion Chromatography Standard (IC), VWR). Limits of detection for $Ca^{2+}$ was 0.01 mg/l. Quality assurance





for IC and determination of pH included continuous analyses of control standards and regular participation in semi-annual
inter-laboratory comparison studies by the World Meteorological Organization (WMO) GAW Programme.

## 3 Results and discussion

### 3.1 Identification of mineral dust layers from thermal-optical analysis and ionic data

The identification of mineral dust layers allows to retrace events with mineral dust deposition during the accumulation period
and is the starting point for mineral dust concentration and deposition assessment. We identified dust-laden samples based on
the temperature dependent change in optical properties of residual material seen in thermal-optical analysis. As described by
Kau et al. (2022), the transmitted laser signal ($\lambda$=660 nm) in the calibration phase was evaluated. Since the temperature range
given previously was not always reached for our set of samples, we now evaluated the change in transmittance observed
between 700 and 450°C. Converted into a temperature dependent attenuation ($ATN_{700-450}$), samples that exceeded a value of
4.0 were classified as containing mineral dust. To account for lower filter loadings, we added high linearity of the relationship
between the temperature and the transmitted laser signal (coefficient of determination, $R^2$, above 0.9) as a criterion. We refer
to this method to identify dust-laden filters as TOA approach.

The TOA approach is compared with a complementary approach based on pH values above 5.6 and $Ca^{2+}$ concentrations above
10 $\mu eq\ L^{-1}$, which was previously used to identify mineral dust layers at the GOK site (Greilinger et al., 2018). In the following
we will refer to this method as IC approach. The following limitations apply: 1) The methods are compared using neighbouring
140  depth profiles taken within one snow pit, but not identical profiles. Hence, differences resulting from uneven terrain or snow
drifts cannot be excluded. Schöner et al. (1997) report coefficients of variation up to 82 % (median up to 24 %) for major ions
except $Na^+$ for single layers of three adjacent profiles within one snow pit at the GOK site. Similar differences can be assumed
for our setting. 2) The depth resolution of the profiles differs. One layer of the IC-profile corresponds to two layers of the TOA
profile. This affects the number of identified layers.
145  The layers identified with the two methods are shown in Figure 1. Different shades correspond to the intensity of the mineral
dust event according to the data. For the IC approach, an additional class using a slightly lower $Ca^{2+}$ concentration of 8.5 $\mu eq\ L^{-1}$
and again a pH value above 5.6 was added. This led to an increased agreement of the two methods in two out of five cases.
Weak events may be overlooked with the previous limit, but also false positive results are possible with the lower limit. For
2018 and 2021, the TOA approach identified less layers than the IC approach. Coloured carbon species may conceal mineral
dust prior to TOA, however, due to its refractive properties the colour can easily be evaluated after the analysis. We found a
reddish colouration on the corresponding filters for 2018 and 2021 after TOA. The loading was too low to trigger the TOA
approach. The colour of filters has previously been related to chemical aerosol composition including mineral dust (Tomza et
al., 2001). Considering layers with coloured filters would match the two approaches in 2018, where the mineral dust layer
observed within the IC-profile is divided in two samples in the TOA-profile. For 2021, two out of four layers identified via
the colouration of the TOA filters would match the IC approach. Still, as the additional two layers cannot be matched, the





overall agreement of the two approaches remains the same. The colour of filters after TOA was checked also for the residual years. No additional layers were found. This shows that the TOA approach may overlook filters with slight colouration but is triggered by most samples coloured reddish after TOA. Few layers, e.g., in 2020, were found only with the TOA approach. Events with high dilution due to strong snowfall may not be recognized with either approach, and depending on the spatial

variability only one may trigger.

Overall, the comparison shows very good agreement of the two approaches considering the aforementioned limitations. Hence, each may be used to identify mineral dust layers. Ideally, both complementary approaches are applied. Further evaluations of mineral dust samples presented in this work refer to the identification with the TOA approach. These samples show sufficient loadings to introduce a bias to TOA.

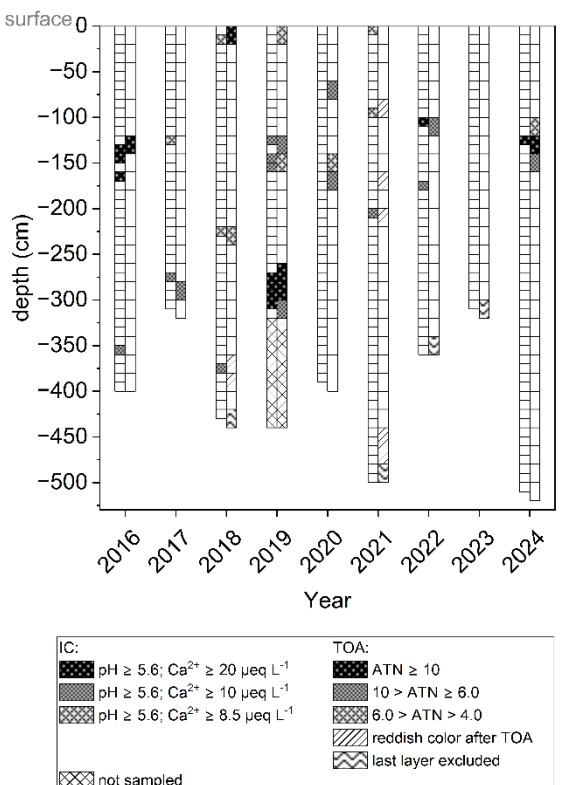


**Figure 1: Identification of mineral dust layers using the TOA approach ($ATN_{700-450} > 4.0$ and $R^2 > 0.9$) and the IC approach (pH value $\geq 5.6$ and $Ca^{2+}$ concentration $\geq 8.5$ or 10 µeq $L^{-1}$, respectively).**



### 3.2 Elemental carbon and water-insoluble organic carbon

#### 3.2.1 Bias introduced by mineral dust: laser correction methods

A suitable laser correction method must be chosen to minimize the bias on OC and EC for samples influenced by mineral dust. We compared results for OC and EC by applying the default Laser/Temp Correction Method in the Calc453 software (Simple Laser/Temp Corr. – Divide by 2) and the linear option (Linear Fit) using the same raw data. The linear option applies a correction that considers temperature dependent changes in the transmittance signal. Thus, the marked increase of the transmitted laser signal observed for dust-laden samples is no longer visible. Due to higher available filter area and hence the

possibility to analyse triplicates, we used weekly PM$_{10}$ filters collected at Sonnblick Observatory. Six filters showed an influence of mineral dust (ATN$_{700-450}$ > 4.0). These samples represent the highest mineral dust loadings at the site between 2019 and 2022. For comparison, six additional filters without influence of mineral dust, collected one or two weeks before or after each mineral dust filter, were analysed.

Results are shown in Figure 2 with error bars corresponding to the standard deviation of triplicates. Changes for OC were

below the uncertainty given by the instrument for both sample groups. The same applies to EC, when samples without mineral dust influence are considered. When mineral dust is observed, the linear option will show higher EC loadings. The increase of the EC signal is ranging from 9.6% to 110% with a median of 51%. This highlights the importance of choosing a suitable laser correction for samples showing a temperature dependent change in optical properties.

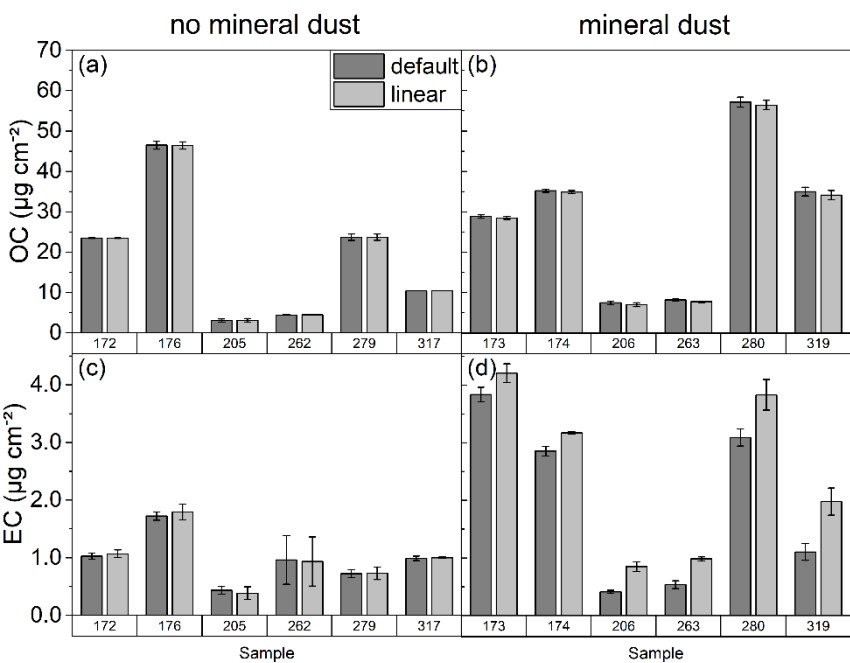

**Figure 2: Comparison of OC and EC loadings using default and linear laser correction setting for 6 PM$_{10}$-filters without mineral dust (a) and (c) and 6 filters with mineral dust influence (b) and (d).**





We apply the linear laser correction to TOA-profile samples influenced by mineral dust to avoid biased results for EC and WinsOC. The PM$_{10}$ dataset suggests that the use of the linear option is justified for all samples, however, we cannot exclude the introduction of a bias if the laser signal is not dominated by mineral dust. Hence, we apply the linear correction only for

mineral dust samples identified with the TOA approach and quantify the effect of the adapted evaluation for EC and WinsOC. As WinsOC concentrations exceed EC concentrations by one to two orders of magnitude, changes become only visible for EC.

For single samples, i.e. the layers affected with mineral dust, median reductions of WinsOC were -0.52 % and ranged up to -2.4 %. For EC the median increase was 63 %, while EC increased up to 2100 %. These extremely high changes for EC were

observed for samples with very high loadings of mineral dust. For them, no EC was detected with the default evaluation, i.e., the laser signal never reached its initial value, and the calculated increase is based on half of the limit of detection. EC concentrations for single layers and changes for layers including mineral dust are shown in Figure 3 exemplary for 2020 and 2024. For entire snowpacks, the correction of bias introduced by mineral dust led to negligible changes for WinsOC with a maximum of -0.25 % and a median of -0.14 %, while the impact for EC is still noteworthy and ranged up to 8.3 % with a

median of 4.7 %. Note that the changes shown here just affect the split between OC and EC determined via the laser signal and TC concentrations will remain unchanged. Carbon introduced by mineral dust, e.g., carbonate or a biofilm adhered to the particles, will still be part of either WinsOC or EC depending on the temperature protocol. The results for EC, i.e., the median increase of 63 % for single samples and the impact on entire snowpacks of up to 8.3 %, underline the importance of identifying samples where mineral dust affects the laser signal and choosing a suitable laser correction method. Further evaluations are

done with the corrected WinsOC and EC data.

### 3.2.2 Concentration and deposition between 2016 and 2024

WinsTC, WinsOC and EC concentrations and depositions are shown in Figure 3. Minimum and maximum concentrations and depositions as well as their average and standard deviation for the winter accumulation periods are given in Table 1. Concentrations are given as the mass of carbon per mass of deposited snow. While WinsTC data cover the whole time period,

WinsOC and EC data are available between 2017 and 2024 only, due to insufficient quality of the laser data in 2016.

**Table 1: Minimum, maximum, average concentrations and depositions and their standard deviations for the accumulation periods 2016-2024 (WinsTC) and 2017-2024 (WinsOC and EC).**

| | Concentration (ng g$^{-1}$) | | | Deposition (mg m$^{-2}$) | | |
|---|---|---|---|---|---|---|
| | WinsTC | WinsOC | EC | WinsTC | WinsOC | EC |
| **Minimum** | 224 | 211 | 8.60 | 241 | 227 | 10.9 |
| **Maximum** | 933 | 917 | 15.5 | 1270 | 1250 | 30.7 |
| **Average** | 459 | 458 | 11.1 | 731 | 743 | 17.8 |
| **Standard deviation** | 204 | 215 | 2.5 | 348 | 355 | 6.5 |



EC accounted for 1.4±0.9 % of WinsTC on average and ranged between 0.39 and 3.0 %. Due to the low share of EC in WinsTC, the courses of WinsOC concentration and deposition resemble those of WinsTC. None of the carbonaceous species show a
trend in the observed period, more data are necessary for robust statistical evaluation. In 2019, sampling had to be stopped due to bad weather and about 72 % of the snowpack regarding depth could be collected. While the lack of these samples leads to an unknown bias for the concentrations of analytes (higher or lower actual concentration possible), it leads to an underestimation of their depositions.

We want to give an overview of WinsOC and EC concentrations and depositions from literature of various sites to compare to
our data. Early data reporting WinsOC and EC concentrations in the Alps were given by Cerqueira et al. (2010). They reported concentrations for WinsOC and EC at Sonnblick in surface snow samples collected between March 2003 and July 2004 ranging between 33-785 µg L$^{-1}$ and up to 12 µg L$^{-1}$, respectively, and average concentrations of 145±174 µg L$^{-1}$ and 5.2±3.7 µg L$^{-1}$, respectively. Contribution of EC to TC was found to be below 6.5 %, which agrees well with the recent data of this work (up to 3.0 %). For Schauinsland (Germany), average EC and WinsOC concentrations in rain and snow were 28±38 µg L$^{-1}$ and
205±266 µg L$^{-1}$, respectively (Cerqueira et al., 2010). More recent data are given for the French Alps, where EC concentrations during the two snow seasons 2016-2017 and 2017-2018 were reported between 0 and 80 ng g$^{-1}$ using coagulant (Tuzet et al., 2020), quite similar to the single samples of the snow profiles shown exemplarily for 2020 and 2024 in Figure 3. Extensive sampling has also been reported for Scandinavia and the Arctic. Mori et al. (2019) report median concentrations of black carbon in snowpack column samples collected between 2012 and 2015 in the Arctic between 1.45 and 10.7 µg L$^{-1}$. Forsström
et al. (2013) report EC column loads for various sites in Scandinavia and the European Arctic, which were collected between 2007 and 2009, spanning from 1.5 to 171.2 mg m$^{-2}$. Zdanowicz et al. (2021) found varying concentrations for WinsOC (<1 to 9426 ng g$^{-1}$) and EC (<1.0 to 266.6 ng g$^{-1}$) in Svalbard at 37 sites between 2007 and 2018. Carbon input varies globally and can reach much higher values. Kaspari et al. (2014) found average black carbon concentrations of 180.0, 24.4 and 1.0 µg L$^{-1}$ at 3 different sites at Mera glacier (Himalaya), with single values spanning up to 3 orders of magnitude within one site. Zhang
et al. (2018) reported black carbon concentrations at 3 regions of the Tibetan Plateau between 1323±242 and 5624±1500 ng g$^{-1}$ in samples collected in December 2014 and November 2015. Our data are comparable with data from the European background and Arctic and provide recent concentrations and depositions of carbonaceous species covering nearly a decade at a central European background site in the Austrian Alps.





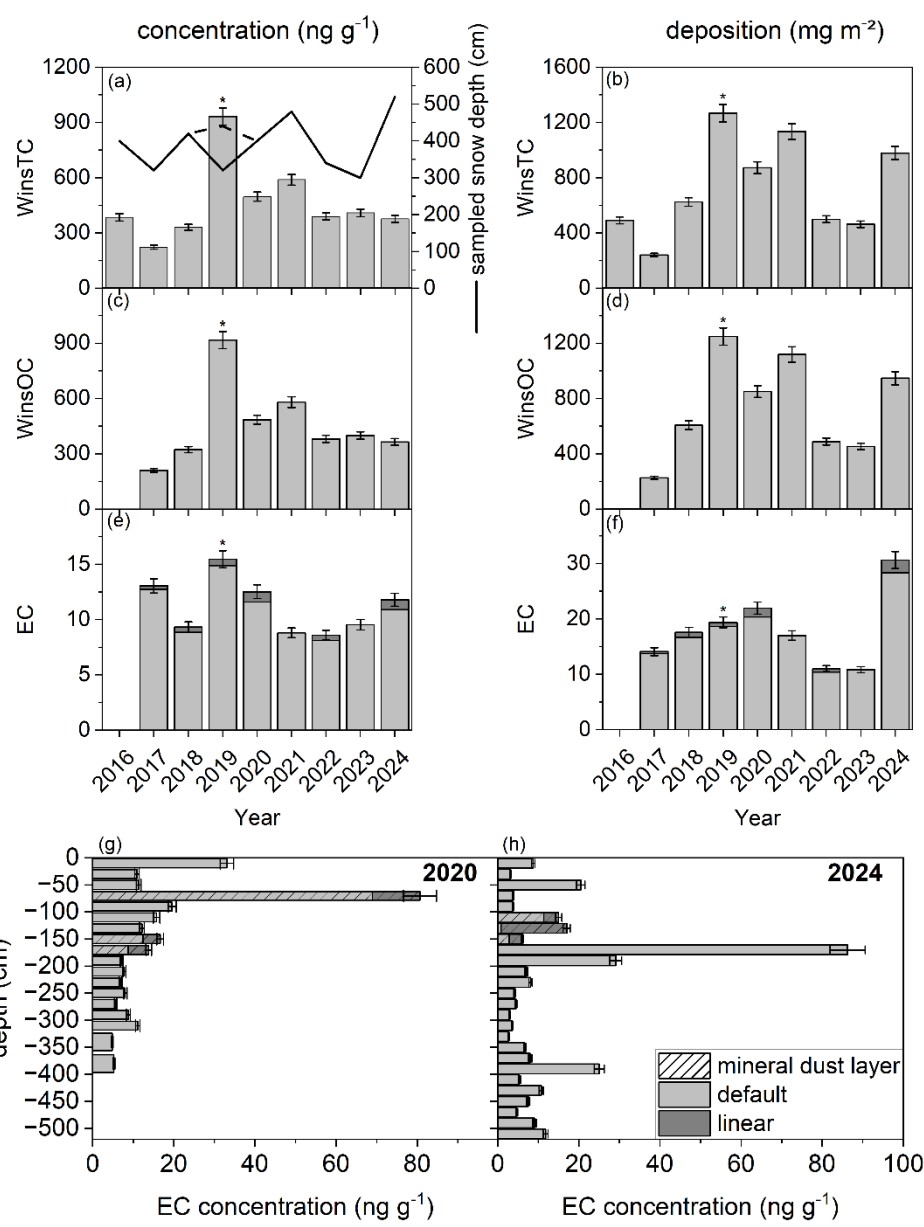

Figure 3: WinsTC concentration (a) and deposition (b) for 2016-2024, concentration of WinsOC (c) and EC (e) and deposition of WinsOC (d) and EC (f) for 2017-2024. Sampled snow depth is shown in (a) as a continuous line. For 2019 (marked with *), sampling had to be interrupted. Dashed line shows actual snow depth for 2019. EC concentration of single layers is shown exemplarily for 2020 (g) and 2024 (h).





## 3.3 Mineral dust

### 3.3.1 Calculation of Fe loading from TOA data

The Fe loading of mineral dust layers was estimated from TOA data as described in Kau et al. (2022). As discussed in Sect. 3.1, the temperature range for the attenuation calculation was adapted in this work, necessitating a new fit to obtain the Fe loading from TOA data. The Fe loading was determined via ICP-OES and related to the $ATN_{700-450}$ values for selected dust-laden filters. As the calculation of a new fit should be based on a maximum of data points and only 7 data points were available for GOK, data of another glacier field close to Sonnblick Observatory, Kleinfleißkees (FLK), were added. Contrary to GOK, sampling there was not conducted annually, but data of mineral dust layers from the years 2017-2018 and 2022-2023 were available. A sum of 14 samples could be used to calculate the fit shown in Eq. 1 and Figure 4. Most available samples showed low Fe loadings. As the highest $ATN_{700-450}$ value was 13, the resulting fit is only applicable between 4.0 and 13 and should be reevaluated when an increased number of samples is available. The differences between the calculated and measured Fe loadings related to the measured Fe loadings for GOK samples spanned between 0.6 and 19% with a median of 13%.

$$\text{Fe loading} = 1.67 * ATN_{700-450} \qquad (1)$$

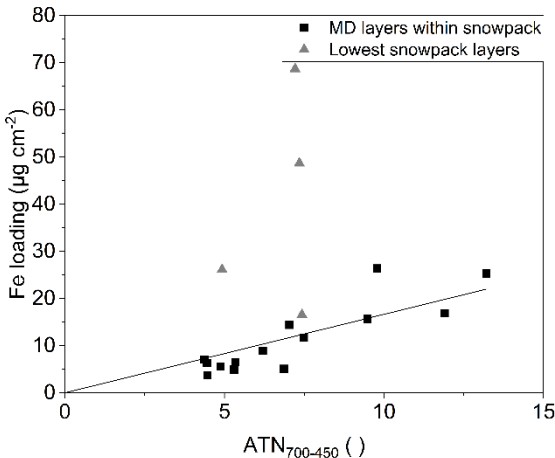

**Figure 4: Relationship of $ATN_{700-450}$ and Fe loading for mineral dust samples within the snowpack. Lowest snowpack layers are shown as triangles.**

Targeting complete sampling, lowest layers may include material deposited prior to the accumulation period, usually containing a considerable amount of dust and carbonaceous particles of varying origin (e.g., cryoconite). This was the case for the TOA-profiles of 2018 and 2021-2023, where the last layer was excluded from the evaluation. Their relationship between $ATN_{700-450}$ and the Fe loading differs from that of samples within the snowpack and shows a steeper slope, see Figure 4. We attribute this to several factors, including mixture with residual material of the last accumulation period, mixture with local dust and possible weathering of last accumulation period's dust.





### 3.3.2 Fe as a proxy to determine mineral dust mass

Identifying mineral dust layers opens the possibility to approximate the mineral dust load via a suitable marker. This would be a useful alternative to the gravimetric determination, which is labour-intensive and cannot easily be applied to quartz fibre filters needed for subsequent TOA to determine EC. We decided to use Fe as a marker, as it is also accessible via TOA (Kau et al. 2022) and would further facilitate an approximation of the dust load within a monitoring program. The approximation of mineral dust from TOA is based on the Fe loading of the filters and the mass fraction of Fe in mineral dust. The average share of Fe in mineral dust was deduced from snow (n=7) and rime (n=1) samples collected at Sonnblick Observatory after intense mineral dust events occurring between 2020 and 2024. For comparison, local dust collected at the surrounding snow-free surfaces in August 2024 was analysed. Detailed description of the sampling and evaluation can be found in Appendix B.

For mineral dust from long-range transport, the share of Fe spanned between 3.0 and 4.5 %, with an average of 3.9 %, a standard deviation of 0.4 % and a median of 4.0 %. In Literature, varying shares of Fe in mineral dust are reported: Di Mauro et al. (2019) found 4.0 % Fe in a Saharan dust sample collected in the Italian Alps in February 2014. Dumont et al. (2023) report a decreasing mass fraction for Fe along the transport path from 11 % in the Pyrenees to 2 % in the Swiss Alps for a mineral dust event in February 2021. This corresponds to higher average total-Fe content of mineral dusts collected close to the source regions, as reported by Lafon et al. (2004), who found 6.3±0.95 % in dusts from the Sahara and 7.8±0.4 % from dusts from the Sahel collected in Niger. The shares of Fe reported for mineral dust samples from long-range transport collected in Europe cover the range found in this work. The median share of Fe in mineral dust mass corresponds to a factor of 25 to estimate the mineral dust loading from the Fe loading.

For the local samples, the share of Fe spanned between 2.1 and 3.4 %, with an average of 2.6 %, a standard deviation of 0.4 % and a median of 2.6 %. The results for both sample groups are shown in Figure 5. The boxplots of the two groups show an overlap, but local samples generally have lower Fe content. Data of additional elements and a higher number of samples would be needed when aiming at a classification between mineral dust from local or remote sources. Further calculations are based on the data set for dust from long-range transport, i.e., a factor of 25 will be applied to approximate mineral dust based on Fe loadings obtained by TOA. Note that these estimations of mineral dust are based on a simple approach assuming relatively stable shares of Fe in mineral dust. The variability of this share is reflected in the resulting uncertainty.





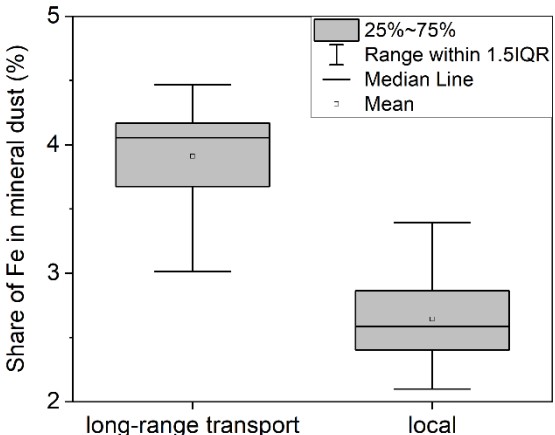

**Figure 5: Share of Fe in mineral dust from long-range transport and local sources. IQR: interquartile range.**

### 3.2.3 Estimation of mineral dust concentration and deposition from TOA data

The resulting mineral dust concentrations and depositions for the years 2016 to 2024 are shown in Figure 6. For the year 2016,
the estimation is based on the Fe loading determined via ICP-OES, as the laser data were of insufficient quality. Uncertainty
includes the variability of the share of Fe in mineral dust (25±3) and spreading of data points around the fit to calculate Fe
loading from $ATN_{700-450}$ (11±7). The first includes variation due to the composition of mineral dust and from the ICP-OES
analysis procedure, while the latter includes variation in composition and variation of the laser signal from TOA. We assume
the error in retention of the quartz fibre filter to be negligible, as the mode of mineral dust particle sizes was between 4.5 and
13 µm (determined with Mastersizer 2000, Malvern Panalytical), which is in good agreement with dust-laden samples
collected in the Italian Alps showing modes of 7.9 and 8.5 µm (Di Mauro et al., 2019). Calculation of the uncertainty of the
mineral dust approximation was based on error propagation and resulted in 65 %. While the uncertainty may seem substantial,
we want to point out that the spatial variation of dust input is expected to be high. Studies analysing spatial variation reported
for various parameters, e.g., dust deposition, ionic input, snow depth, and scales, ranging from within one snow pit to larger
scales, show high variability, even if often not quantified (e.g., Rohrbough et al., 2003; Dumont et al., 2023; Di Mauro et al.,
2015; Rohde et al., 2023). In the smallest scale, within one snow pit, coefficients of variations ranged up to 82 % with a median
up to 24 % for major ions excluding $Na^+$ at the GOK site (Schöner et al., 1997). An uncertainty of this magnitude can be
expected and can never be reduced. The method introduced here allows the approximation of mineral dust as an additional
parameter deduced from EC monitoring, as no change in the analytical routine is necessary.



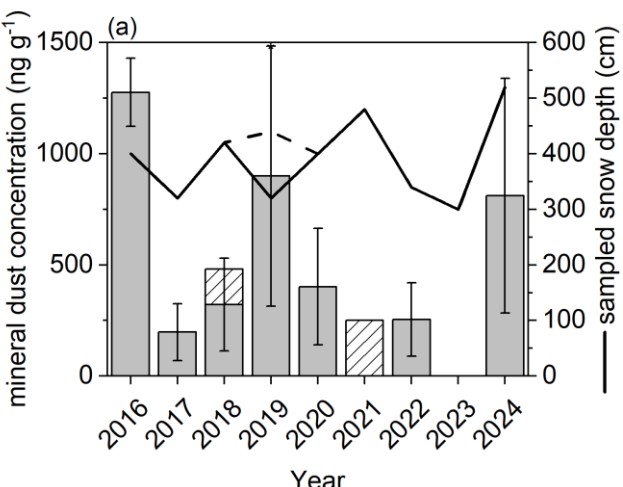
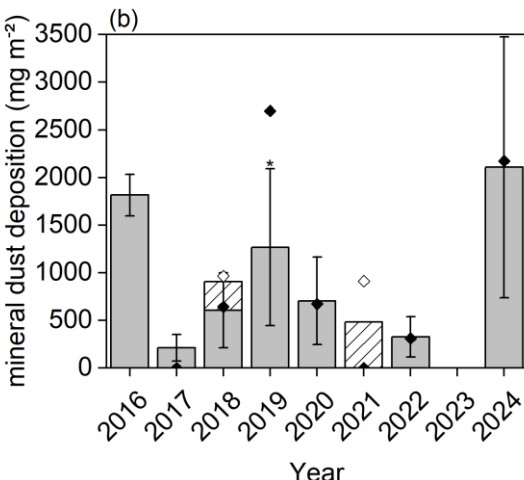


**Figure 6: (a) Sampled snow depth (continuous line), estimated mineral dust concentration and (b) deposition (bars) for 2016-2024. For 2019 (marked with \*), sampling had to be interrupted. Dashed line shows actual snow depth for 2019. Hatched areas correspond to filters, which were not identified as containing mineral dust but showed reddish colour after TOA. Particle mass on mineral dust filters determined by gravimetry is shown as black diamonds in (b); white diamonds consider also dashed mineral dust layers.**

Mineral dust concentrations ranged up to 1300 ng $g^{-1}$. Highest mineral dust concentrations were found for the years 2016, 2019 and 2024. Note that the incomplete sampling of the snow cover in 2019 leads to an unknown bias in the concentration, while deposition may be underestimated. This was already discussed in Sect. 3.1.2. Assuming no deposition of mineral dust in the lower part of the snowpack and a similar density of snow layers, mineral dust concentration may be decreased by 28 %. The concentration of mineral dust would still be within the top 3 years in the researched period. For single layers, highest mineral

dust concentrations were found in 2016 and 2024 and ranged up to 25 and 16 µg $g^{-1}$, respectively. This comes close to concentrations of strong mineral dust events observed in the Italian Alps, where mean concentration of surface snow from six spots was 54.5 µg dust (g snow)$^{-1}$ in 2016 (Di Mauro et al., 2019).

Deposition of mineral dust ranged up to 2100 mg $m^{-2}$. Mineral dust deposition was highest in the same years as the concentration, however, 2024 exceeding 2016 and 2019. Mineral dust deposition in 2019 can only be underestimated and may

be the second highest or highest year regarding mineral dust input. For the February event in 2021, Dumont et al. (2023) analysed samples collected in the Pyrenees, French Alps and Swiss Alps and found depositions between 0.2 and 58.6 g $m^{-2}$. Deposited mass decreased along the transport path and led to mean values of 21.2, 7.2 and 3.5 g $m^{-2}$ for the aforementioned sites, respectively. Meinander et al. (2023) modelled deposition of Saharan dust in February 2021 over Finland. Maximum total wet deposition of dust was between 500 and 1000 mg $m^{-2}$, while their simulations show how dust transport bypassed

Austria. This is consistent with our findings that no intense mineral dust layer corresponding to this transport event was found at the GOK site. The orders of magnitude in our work agree with these findings, especially for 2016, where only one layer dominated the mineral dust deposition. Compared to most studies, our results do not focus on deposition of mineral dust



resulting from single events but represent the mineral dust input throughout the entire winter accumulation period in nine consecutive years.

For filters showing reddish colour after TOA but not triggering the TOA approach, mineral dust was estimated from elemental data and is also shown in Figure 6. Considering these filters would lead to an increase of mineral dust concentration and deposition by 160 ng g$^{-1}$ and 300 mg m$^{-2}$ (50 %) for 2018 and 250 ng g$^{-1}$ and 490 mg m$^{-2}$ for 2021, respectively. For 2021, four separate layers were evaluated. Note that each layer would contribute only little to the mineral dust input. The two layers for 2018 are concentrated in one layer for the IC-profile, while they are separated in the TOA-profile. They are considered

"overlooked" and should be included for future evaluations. For quality assurance, checking the filters' colour after TOA is recommended.

Gravimetric data of insoluble particles are available for 2017-2022 and 2024. However, an alternative to gravimetric determination of mineral dust is desirable, as several challenges occur: Care must be taken to take the mass of the filters before and after filtration at comparable conditions, i.e., comparable level of humidity, as particle loadings are usually low and can

easily be biased. Fibres of the filter material are lost during handling (visible as white residue in the filtration device), leading to an underestimation of particle mass. Hence, most blanks show negative masses after filtration of ultrapure water, which necessitates a correction of the filter loadings. The variability of the losses observed for blanks leads to high limits of detection. Due to the low particle loadings at our site, only 36 out of 131 samples corresponding to 27 % of samples were above these limits of detection. Still, most mineral dust samples exceeded the limits of detection (87 %). Two mineral dust filters were

below the limit of detection (one from 2017 and 2018 each, corresponding to 100 and 50 % of mineral dust layers of these years). For 2017, 2018 and 2020 (no data available for one sample), gravimetric data presented underestimates actual particle mass of the mineral dust layers. Furthermore, gravimetry by itself is not a reliable criterion to identify mineral dust samples. In Figure 6, we show the deposited particle mass from the gravimetric evaluation of the mineral dust filters for 2017-2022 and 2024 assuming that all insoluble particles can be attributed to mineral dust. Overall, our newly presented method agrees well

with gravimetric data. If an appropriate estimate for the share of Fe is available, a mineral dust estimate from archived TOA data is accessible.

## 4 Conclusions

We report continuous data of water-insoluble total carbon (WinsTC), water-insoluble organic carbon (WinsOC) and elemental carbon (EC) for snowpacks sampled between 2016 and 2024. A linear laser correction was applied to evaluate samples

containing mineral dust to minimize the bias caused by the temperature dependent changes in optical properties. Changes due to the adapted evaluation showed negligible influence on WinsOC for single samples and entire snowpacks, but median increases for EC of 63 and 8.3 %, respectively. This highlights the necessity to choose a suitable laser correction for samples including mineral dust in thermal-optical analysis (TOA). Average concentrations for WinsTC, WinsOC and EC were



459±204 ng g$^{-1}$, 458±215 ng g$^{-1}$ and 11.1±2.5 ng g$^{-1}$ and depositions were 731±348 g m$^{-2}$, 743±355 mg m$^{-2}$ and

17.8±6.5 mg m$^{-2}$, respectively.

We identified mineral dust layers in the snowpacks using two complementary approaches based on TOA data or ion chromatography data combined with the pH. They agreed very well and we conclude that each may be used. The median share of Fe in mineral dust from long-range transport deposited at Sonnblick Observatory between 2020 and 2024 was 4.0 %. Using this value, we estimated the mineral dust concentration and deposition based on TOA data. Concentration in the snowpacks

ranged up to 1300 ng g$^{-1}$ and deposition up to 2100 mg m$^{-2}$. The results agree well with the gravimetric data of these layers and strongest events with data reported in literature. The approximation of mineral dust from TOA data is applicable for monitoring assuming a rather stable share of Fe in mineral dust mass. Identification and approximation can be retrospectively applied to data of samples containing mineral dust, as no change in analytical routine is necessary.

**Appendix A: Use of coagulant in melted snow samples?**

In liquid filtration, EC filtration efficiency of quartz fibre filters is still discussed. In literature, different samples (standards or precipitation) were analysed using a series of filters or via comparison with other methods and reported filtration efficiencies for EC ranged from 10 to 92% (e.g., Ogren et al. (1983), Torres et al. (2014), Hadley et al. (2008), Lim et al. (2014)). We did not assess the EC filtration efficiency for our samples, but assessment for similar sample preparation and types gave an undercatch of 22% (Zdanowicz et al., 2021). Using reference materials (oxidized black carbon), addition of coagulant

(NH$_4$H$_2$PO$_4$) showed an increase of the filtration efficiency from 5 to 95% (Torres et al., 2014). For real snow samples including aged carbon, filter efficiency was higher and addition of the coagulant showed a smaller effect on EC filtration efficiency (increase by a factor of 1.45; Kuchiki et al., 2015). Only a small number of publications applying the method are available (e.g., Tuzet et al.,2020; Thind et al., 2021). We added NH$_4$H$_2$PO$_4$ to a limited number of samples and compared it to untreated aliquots to assess the applicability for our background site. Addition of the coagulant led to a decrease in pH

(ultrapure water: 5.6 to 4.4; sample containing mineral dust: 8.2 to 4.6), which can alter the sample composition, e.g., by removing carbonate carbon. We observed changes in the signal of the flame ionisation detector in various OC and EC fractions (positive and negative) and obtained blanks with doubled TC loadings. In agreement with Kuchiki et al. (2015), no influence of an aqueous solution of the substance pipetted on a clean filter on the laser signal was observed. However, no automatic split point could be set for any replicate of a sample not containing mineral dust, while the untreated sample showed EC. This

suggests that the coagulant can interact with compounds of real snow samples leading to changes in optical properties. Due to the influence on sample composition and analysis, and the lack of datasets from background regions applying the method for comparison, we decided not to modify our samples by adding coagulant. Hence, we expect an underestimation of EC. Uncertainties regarding homogeneity of the filter loading do not apply in our case, as the entire loaded area was analysed, and we do not expect losses to the container walls during sample preparation based on Forsström et al. (2009).



## Appendix B: Sampling, processing and evaluation of dust from long-range transport and local sources


Dust from long-range transport was obtained from snow and rime collected at Sonnblick Observatory after intense transport events. Samples were melted in a beaker using a microwave (600 W for 1 min, homogenized, repeated until fully melted) and transferred to closed plastic bottles. The particles were left to settle for 24 h and the supernatant was decanted. The settled dust was transferred to an evaporating dish with a small amount of ultrapure water and was put on a hotplate at around 125°C (not

boiling) to remove most of the liquid. The moist dust was dried at 115°C in a drying oven over night.

Dust from local sources was collected on snow-free surfaces close to Sonnblick Observatory (n=4) using a brush and for one sample a hand-held vacuum cleaner. Visual inspection allowed to select samples resembling the dust from long-range transport obtained from surface snow and rime in terms of particle size. All samples were collected at surfaces without visible signs of vegetation.

Replicates of each sample (each about 12 mg) were digested, and Fe was quantified via ICP-OES. For half of the samples from long-range transport (overall n=8), triplicate measurements could be done, while for the other half only duplicate measurements were possible due to limited sample mass. To avoid unbalanced influence of the samples due to varying numbers of replicates, the average of the replicates was used for each sample in the subsequent evaluation. This was valid, as variation in replicates was small (triplicates: relative standard deviation; duplicates: half of the difference related to the average) and ranged between

0.1 and 4.1% with a median of 1.3%.

For the local samples (n=4), triplicate analyses were done. Relative standard deviation ranged from 5.4 to 20% with a median of 9.9%. We contribute the high variation to inhomogeneity of the samples. Due to the even number of replicates, low overall number of samples and high variation within one sample, each result was considered by itself for the evaluation of the Fe share in the dust samples.

**Data availability**

Data used in this work will be uploaded to TU Wien Research Data and the doi will be added here.

**Author contribution**

**DK:** Conceptualization, Data curation, Formal analysis, Investigation, Methodology, Validation, Visualization, Writing (original draft preparation), Writing (review and editing). **MG:** Data curation, Formal analysis, Writing (review and editing).

**AV:** Investigation, Writing (review and editing). **JB:** Investigation, Writing (review and editing). **LK:** Investigation, Writing (review and editing). **AKG:** Conceptualization, Methodology, Resources, Writing (original draft preparation), Writing (review and editing).



**Competing interests**

The authors declare that they have no conflict of interest.

**Acknowledgements**

We thank Gerd Mauschitz from the Research Group for Particle Technology, Recycling Technology and Technology
Assessment from TU Wien for the opportunity to analyse particle size distributions. Special thanks go to a group of students
who helped with chemical analysis of samples. Snow sampling is nowadays financed by general resources from the GeoSphere
Austria. Sampling is supported by colleagues from the section climate monitoring and cryosphere and from the Sonnblick
Observatory. We thank www.foto-webcam.eu for permission to use a part of a photo from "Goldbergkees 2" for the graphical
abstract. The authors acknowledge TU Wien Bibliothek for financial support through its Open Access Funding Programme.

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
