# Peer review of "Light-absorbing snow impurities: Nine years (2016-2024) of snowpack sampling close to Sonnblick Observatory, Austrian Alps"

_EGUsphere, 2025_

## Referee Comment (RC1)

**Review Kau et al., 2025**

**Light-absorbing snow impurities: Nine years (2016-2024) of snowpack sampling close to Sonnblick Observatory, Austrian Alps**

Kau et al. analyze water-insoluble organic carbon and elemental carbon in snow samples collected from the eastern Alps, near the Sonnblick Observatory, between 2016 and 2024. This study emphasizes the interference caused by dust during thermal optical analysis (TOA), which can lead to a significant underestimation of elemental carbon (EC). The authors introduce a laser correction method that offers two main advantages: it enables a more accurate quantification of EC and facilitates the detection and estimation of dust concentration. The detection of dust aligns with its presence identified through pH and calcium measurements. Additionally, although gravimetric analysis can quantify dust for a limited number of samples, it supports the dust estimation derived from the TOA approach.

The main limitation of the present manuscript is the lack in explanation about the procedure employed for TOA correction. This makes it difficult for the redear to replicate the same approach. I recommend the manuscript for publication in ACP after major revisions.

**Main comments**

Section 2.2.1 How do you think the non-homogeneous distribution of the LAPs on the filter affects the quantification of OC and EC? In particular, are you able to estimate the related uncertainty?

Line 128-136. It would benefit readers if the authors could provide more details about the procedure used to detect the dust layer based on TOA. For instance, does '4' refer to the change in transmittance during the cooling phase from 700 to 450 °C? If so, it would be clearer to denote it as delta ATN (also in Figure 1). The statement "Since the temperature range previously mentioned was not always reached for our set of samples" suggests that a different thermal procedure was used in prior analyses. Additionally, please clarify why the color change observed during the cooling phase indicates the presence of mineral dust.

Line 172. The authors should specify whether the "Laser/Temp Correction Method in the Calc453" refers to the default correction, which does not account for changes in transmittance with temperature due to the presence of dust. Is this the same as the "default method" indicated in Figure 2? Moreover, it would be helpful if the authors could elaborate on the correction based on the linear fit in the methods section, as this information is crucial for ensuring the replicability of the approach.

Line 189. Please explain why this method can only be applied to samples where the optical signal is predominantly influenced by dust, and how this can be inferred. While you can estimate dust presence through calcium measurements or pH data, this does not provide information on the content of elemental carbon (EC) and the relative contributions of these two components to the optical properties of the sample. Additionally, if dust is negligible, I assume

that applying the linear fit correction would not result in any significant changes to the results, as illustrated in Figure 2. Is this assumption correct?

Line 260-265: The differing behavior of the snowpack with mineral dust (MD) and the lowest snowpack layers shown in Figure 4 suggests that MD from various sources exhibits distinct optical responses. Is this due to the fact that optical properties are not solely determined by iron content? Could the authors elaborate on this point further? Would the authors recommend validating the linear fit method with field measurements to ensure its applicability in different environments or regions?

Fig 5. Presenting the statistical distribution of eight measurements using a box-whisker plot can be misleading, as this plot summarizes a dataset consisting of only eight data points with just six numbers. I would suggest simply reporting the median and providing the range in brackets within the text.

Page 15 discusses previous studies that recommended quantitatively measuring dust using polycarbonate filters (Kuchiki etal., 2015). Why did the authors choose not to pursue this method? Did they plan to use the same filter for both gravimetric measurements and thermal optical analysis? Furthermore, does the comparison between thermal optical analysis (TOA) and gravimetric dust quantification suggest that the TOA approach should be applied to all reddish filters, and should the defined threshold for triggering measurements be modified?

In the conclusions section, while summarizing the study's results, the authors should also emphasize the caveats and limitations of the described method, as well as the implications of their findings, as outlined in the author guidelines (https://www.atmospheric-chemistry-and-physics.net/policies/guidelines_for_authors.html).

**Technical corrections**

Line 44. Please add references to support this sentence:" and is commonly applied to filters loaded with insoluble particles from snow samples."

Line 46. "A comprehensive discussion of the impact of this bias on elemental carbon results in seasonal snow covers is still lacking." This sentence is correct when referring specifically to thermal optical method. I suggest revising this sentence to clarify it refers to one specific measurement technique.

Pag9. I suggest reporting the concertation of OC and EC from previous studies in a table. This would make it easier for the reader to compare the previous results with the ones reported in this work.

Line 296-297: please define uncertainty units

Line 299. Please specify that the indicated size range refers to dust particles in snow (that might have been subject to deposition post-processing). In facto atmospheric dust particles show a smaller diameter range (SCHWlKOWSKI et al., 1998).

**References**

Kuchiki, Katsuyuki, Teruo Aoki, Masashi Niwano, Sumito Matoba, Yuji Kodama, and Kouji Adachi. "Elemental carbon, organic carbon, and dust concentrations in snow measured with thermal optical and gravimetric methods: Variations during the 2007–2013 winters at Sapporo, Japan." Journal of Geophysical Research: Atmospheres 120, no. 2 (2015): 868-882.

Schwikowski, M., Seibert, P., Baltensperger, U. and Gaggeler, H.W., 1995. A study of an outstanding Saharan dust event at the high-alpine site Jungfraujoch, Switzerland. Atmospheric Environment, 29(15), pp.1829-1842.

---

## Author Comment (AC1)

**Answer to Referee Comment of Anonymous Referee #1**

We thank the anonymous referee for their time to review our manuscript and the positive feedback. We appreciate the valuable input, which allowed us to improve our manuscript.

We structured our answer according to the referee comment, i.e., (1) Main comments and (2) Technical corrections, and give the comment in bold letters while our answers are given in common formatting. Changes in the text are given within quotation marks in italics.

**(1) Main comments**

**Section 2.2.1 How do you think the non-homogeneous distribution of the LAPs on the filter affects the quantification of OC and EC? In particular, are you able to estimate the related uncertainty?**

A non-homogeneous distribution of LAPs on the filter will definitely affect the quantification of OC and EC if only parts of the filter are analysed. To prevent this influence, we concentrated the water-insoluble particles of our filters on circular spots with a diameter of 16 mm, cut the loaded area using a punch of 17 mm and analysed both halves of the filter. Thus, all particles retained by the filter were analysed between 2017 and 2024.

For the year 2016, punches with a diameter of 15 mm were made. This is less than the loaded area and inhomogeneities could affect the result, as discussed in the manuscript. Based on a separate dataset of snow samples collected spatially and analysed temporally close to the samples described within the current manuscript, an average underestimation of TC of 18% with a standard deviation of 9% was found for samples with visible inhomogeneities. Based on single samples ($n=9$) the underestimation spanned between 7 and 32%. This uncertainty already includes the uncertainty of the analysis.

We agree that a quantification of this effect improves the manuscript and consequently performed the following changes:

We added the following information in Section 2.2.1:

"*An underestimation of water insoluble total carbon between 7 and 32% was found for comparable samples (n=9) with an average of 18% and a standard deviation of 9%. Thus, we report results for 2016 with an increased uncertainty of 50%.*"

We consequently increased the error bar of WinsTC in Figure 3 to 50%.

For samples of 2016 containing dust, no inhomogeneities were visible. Thus, the error bar for MD in 2016 in Figure 6 was not increased.

**Line 128-136. It would benefit readers if the authors could provide more details about the procedure used to detect the dust layer based on TOA. For instance, does '4' refer to the change in transmittance during the cooling phase from 700 to 450 °C? If so, it would be clearer to denote it as delta ATN (also in Figure 1). The statement "Since the temperature range previously mentioned was not always reached for our set of samples" suggests that a different thermal procedure was used in prior analyses. Additionally, please clarify why the color change observed during the cooling phase indicates the presence of mineral dust.**

We agree that the text was not clear. Consequently, we adapted the paragraph as given below.

Still, we want to answer the referee's questions first: Yes, "4" refers to the change in transmittance during the cooling phase and we agree to denote it as $\Delta ATN_{700-450}$, as it is the difference between the attenuation at 700 and 450°C (without requiring the incident intensity of the transmitted laser signal). To avoid confusion, we stuck with $ATN_{700-450}$ in the answers to both referees except for this comment.

All samples were analysed with the same thermal procedure, which defines the temperature setting during the heating period. Calibration takes place after the actual analysis, when heating is stopped and a fan helps with cooling. So the actual analysis, which can be done according to different thermal procedures, is over and the oven just needs to cool down, which is supported by a small fan. This time period is utilized to do the calibration. Data acquisition is continued just long enough to catch the calibration peak. Maintenance at the instrument led to small changes in the position of the insulation material and the efficiency of cooling. Consequently, the oven no longer cooled down to 400 °C before data acquisition stopped. To avoid similar problems in the future, we reduced the temperature interval and evaluated the temperature range between 700 °C and 450 °C only. The color change observed during cooling is most likely induced by hematite, a phenomenon described by Yamanoi et al. (2009) for the analysis of α-$Fe_2O_3$ powders. Still, we did not manage to link the effect directly to the hematite content of the samples, but to the overall iron content. More details are given in Kau et al. (2022).

*"The identification of mineral dust layers allows to retrace events with mineral dust deposition during the accumulation period and is the starting point for mineral dust concentration and deposition assessment. We identified dust-laden samples based on the temperature dependent change in optical properties of residual material seen in thermal-optical analysis. If mineral dust is present this refractory material remains on the filter and shows a reddish color due to the presence of iron oxides. As described by Kau et al. (2022), the transmitted laser signal (λ=660 nm) was evaluated in the calibration phase. At this point the actual analysis of carbonaceous compounds is already finished. Oven and filter sample are just cooling down, but data is still logged to record the calibration peak. Kau et al. (2022) evaluated a temperature range between 700 and 400°C. Due to small adjustments at the fan which supports cooling, the lower temperature, i.e., 400°C, given previously was not always reached at the end of the calibration phase for our set of samples. Thus, we now evaluated the change in transmittance observed between 700 and 450°C using the transmitted laser signal $I_{700}$ and $I_{450}$, respectively. Converted into a dimensionless temperature dependent attenuation ($\Delta ATN_{700-450}$, see Eq. (1)), samples that exceeded a value of 4.0 were classified as containing mineral dust. For clarification, $\Delta ATN_{700-450}$ corresponds to $ATN_{700-400}$ defined in Kau et al. (2022), however, using the intensity of the transmitted laser signal at 450°C. To account for lower filter loadings, we added high linearity of the relationship between the temperature and the transmitted laser signal (coefficient of determination, $R^2$, above 0.9) as a criterion. We refer to this method to identify dust-laden filters as TOA approach."*

$$\Delta ATN_{700-450} = 100 * \ln \left( \frac{I_{450}}{I_{700}} \right) \tag{1}$$

**Line 172. The authors should specify whether the "Laser/Temp Correction Method in the Calc453" refers to the default correction, which does not account for changes in transmittance with temperature due to the presence of dust. Is this the same as the "default method" indicated in Figure 2? Moreover, it would be helpful if the authors could elaborate on the correction based on the linear fit in the methods section, as this information is crucial for ensuring the replicability of the approach.**

Yes, they are the same. In the revised version of the manuscript, we consistently refer to the two Laser/Temp Correction Methods as either "default option" or "linear option" to make the text more comprehensible. Changes were made in Section 3.2.1 as well as the label of Figure 2.

As indicated in the text, both options for the Laser/Temp Correction Method are made available by the manufacturer in the Calc software. To ensure the replicability of the approach, the software version (Calc453) and the exact option names ("Simple Laser/Temp Corr. – Divide by 2" and "Linear Fit", respectively) are given in the text. Unfortunately, no description of the Laser/Temp Correction Methods is provided by the manufacturer. Own postprocessing of the raw data indicates that the "linear option" corrects for the linear change of the transmittance signal in the calibration phase.

**Line 189. Please explain why this method can only be applied to samples where the optical signal is predominantly influenced by dust, and how this can be inferred. While you can estimate dust presence through calcium measurements or pH data, this does not provide information on the content of elemental carbon (EC) and the relative contributions of these two components to the optical properties of the sample. Additionally, if dust is negligible, I assume that applying the linear fit correction would not result in any significant changes to the results, as illustrated in Figure 2. Is this assumption correct?**

The assumption of Anonymous Referee #1 regarding no significant changes using the linear option for the Laser/Temp Correction Method for samples with negligible amounts of mineral dust is comprehensible. Still, we cannot generally suggest using the linear option for all sample types and sites, where various particle types may occur. For samples containing mineral dust, the dust will have a dominant influence on the laser signal. The use of the linear option reduces this bias and is therefore needed. For all other cases it is preferable to stick to the default option, which is the commonly used method. This also increases comparability between different data sets of EC obtained by TOA. Thus, we stick to our previous recommendation and did not make changes in the text.

**Line 260-265: The differing behavior of the snowpack with mineral dust (MD) and the lowest snowpack layers shown in Figure 4 suggests that MD from various sources exhibits distinct optical responses. Is this due to the fact that optical properties are not solely determined by iron content? Could the authors elaborate on this point further? Would the authors recommend validating the linear fit method with field measurements to ensure its applicability in different environments or regions?**

Yes, optical properties are not exclusively connected to the Fe content. We want to point out that Fe can be present in various compounds, which feature different properties. Mineral dust includes Fe containing minerals exhibiting a temperature dependent change in optical properties, e.g., hematite. In contrast, particulate matter samples collected in railway tunnels do not show this behaviour despite their high Fe loadings. Here, Fe is present in other forms, e.g., Fe metal. This was discussed in Kau et al. (2022).

Attenuation depends on various sample parameters including particle concentration, size, shape and refractive index (e.g., Baker and Lavelle, 1984 and references therein). As discussed for the lowest layers shown in Figure 4, we attribute the differing relationship of the lowest layers to several factors, which include mixture with residual material of the last accumulation period, mixture with local dust and possible weathering of last accumulation period's dust. These would affect the exhibited attenuation of the dust.

An influence of different source regions may lead to differences in composition and particle size distribution of mineral dust, possibly affecting the fit between the attenuation and the Fe loading. This can be the case for different sampling sites as well, as aerosol properties can change depending on the duration or distance of transport. Before applying the fit calculated for our site for different regions, we advise validation. To emphasize this, we added the following statement in Section 3.3.1:

*"Note that the relationship between $ATN_{700-450}$ and the Fe loading may differ for samples collected at other sites. Thus, the fit obtained here should be validated prior to application."*

**Fig 5. Presenting the statistical distribution of eight measurements using a box-whisker plot can be misleading, as this plot summarizes a dataset consisting of only eight data points with just six numbers. I would suggest simply reporting the median and providing the range in brackets within the text.**

We agree that the use of boxplots in this case can be misleading. We decided to use the boxplots in the submitted version of the manuscript, as both the variation of the data points in the two groups and measures of central tendency, i.e., median and average values, can be directly compared.

To consider the comment of Anonymous Referee #1, we included the data points in the boxplots. Thus, the number and distribution of samples in each group are easily visible and possible misinterpretation of the graph is avoided. The positive aspects, i.e., visual comparison of the groups, is still kept. The new Figure 5 is shown here:

[Figure]

**Page 15 discusses previous studies that recommended quantitatively measuring dust using polycarbonate filters (Kuchiki et al., 2015). Why did the authors choose not to pursue this method? Did they plan to use the same filter for both gravimetric measurements and thermal optical analysis? Furthermore, does the comparison between thermal optical analysis (TOA) and gravimetric dust quantification suggest that the TOA approach should be applied to all reddish filters, and should the defined threshold for triggering measurements be modified?**

Yes, due to the limited amount of sample and low analyte concentrations one filter needs to be used for the gravimetric measurements and thermal-optical analysis. Even if higher sample masses were available, preliminary tests showed that melting the sample, homogenizing it and splitting it to several filters does not give satisfactory reproducibility. Taking another depth profile for gravimetry would introduce the uncertainty of spatial variability, as discussed in Section 3.1 for the TOA- and IC-profile. Polycarbonate filters are not suitable for thermal-optical analysis, thus, gravimetry had to be conducted with quartz fibre filters. This is now clarified in the manuscript in Section 3.3.3:

*"Gravimetric data of insoluble particles are available for 2017-2022 and 2024 based on the quartz fibre filters subsequently used for TOA."*

The second question addresses the sensitivity of the TOA approach, which is capable of identifying mineral dust loaded samples if $ATN_{700-450}$ exceeds 4.0. Our data do not support a reduction of the threshold, as we observed poor correlation between the transmittance and the temperature ($R^2 < 0.9$, usually around 0.98 for samples with mineral dust) resulting in random $ATN_{700-450}$ values from noise. Thus, we cannot quantify small dust loadings. The TOA instrumentation is not optimized for the assessment of mineral dust, but an interference is utilized to assess an additional parameter. In our study covering nine years, we found coloured filters with mineral dust loadings low enough not to

trigger the TOA approach in two years only. In 2018, the division of a mineral dust layer during sampling led to two samples with lower mineral dust concentrations each ($ATN_{700-450} < 4.0$). While no interference on the OC and EC results is expected, they would be overlooked in the mineral dust approximation if a critical check of the colour of the filters after TOA was dismissed. The same is true for the filters of 2021, where four separate layers showed low mineral dust loadings. Each of these filters would contribute only little to the mineral dust approximation, but multiple filters with low loadings add up. As we recommended checking the filters' colour after TOA for quality assurance in the submitted version of the manuscript (line 340-341) already, we did not make changes.

**In the conclusions section, while summarizing the study's results, the authors should also emphasize the caveats and limitations of the described method, as well as the implications of their findings, as outlined in the author guidelines (https://www.atmospheric-chemistry-and-physics.net/policies/guidelines_for_authors.html).**

We appreciate the thorough review of the manuscript regarding the author guidelines. The manuscript was submitted to The Cryosphere and we rechecked the information given for this journal. In contrast to Atmospheric Chemistry and Physics, we could not find requirements for the conclusions section. As we still find our conclusions section suitable, we did not make changes to the text.

**(2) Technical corrections**

**Line 44. Please add references to support this sentence:" and is commonly applied to filters loaded with insoluble particles from snow samples."**

The text now reads:

*"The latter is the reference method for the quantification of elemental carbon in ambient air samples (DIN e.V., 2017) and is commonly applied to filters loaded with insoluble particles from snow samples (e.g., Zdanowicz et al., 2021; Tuzet et al., 2019; Forsström et al., 2013)."*

As all added references were cited at other positions in the submitted manuscript already, no changes were made to the References section.

**Line 46. "A comprehensive discussion of the impact of this bias on elemental carbon results in seasonal snow covers is still lacking." This sentence is correct when referring specifically to thermal optical method. I suggest revising this sentence to clarify it refers to one specific measurement technique.**

The text now reads:

*"The co-occurrence of mineral dust leads to an interference in the quantification of elemental carbon in thermal-optical analysis and adapted evaluation of measurement data was proposed without quantifying the impact on elemental carbon concentrations (Wang et al., 2012; Gul et al., 2018)."*

**Pag9. I suggest reporting the concertation of OC and EC from previous studies in a table. This would make it easier for the reader to compare the previous results with the ones reported in this work.**

We agree that a table is a preferred type of presentation for data giving an overview. However, the data referred to within our work comprises concentration and deposition data presented in different ways (ranges, averages, medians) by authors for different sample types at various sites and points in time. As the resulting table would not be clear, we did not include the data in a table.

**Line 296-297: please define uncertainty units**

We are thankful for the comment. While the first parameter describing the share of Fe in mineral dust is dimensionless, the unit (%) for the second parameter was missing.

The share of Fe in mineral dust is based on the mass fraction of Fe in mineral dust (median: 4%). For the calculation of uncertainty, its reciprocal value (1/0.04 = 25) and respective standard deviation is used, as the factor of 25 is used to estimate the mineral dust loading from the Fe loading (line 282-283). To consider the spread around the fit to calculate Fe from $ATN_{700-450}$, we calculated the difference between the calculated and measured Fe loadings and related it to the measured Fe loadings (line 254-255). The average and standard deviation in % were used for error propagation.

We added the unit of the latter parameter in Section 3.3.3. To increase comprehensibility, we added the average of this parameter (11%) in Section 3.3.1 (previously only the span and the median were given).

**Line 299. Please specify that the indicated size range refers to dust particles in snow (that might have been subject to deposition post-processing). In facto atmospheric dust particles show a smaller diameter range (SCHWIKOWSKI et al., 1998).**
We included the comment. The text now reads:

*"We assume the error in retention of the quartz fibre filter to be negligible, as the mode of deposited mineral dust particle sizes in snow was between 4.5 and 13 µm (determined with Mastersizer 2000, Malvern Panalytical), which is in good agreement with dust-laden snow samples collected in the Italian Alps showing modes of 7.9 and 8.5 µm (Di Mauro et al., 2019)."*

We want to point out additional changes, which were implemented independent of the referees' comments:

For a more accurate description, we changed "share of Fe in mineral dust" to "Fe mass fraction in mineral dust".

The given WinsTC concentration of the snowpack collected in 2016 slightly differed from the actual concentration (385 and 345 ng $g^{-1}$, respectively) due to a calculation error. We apologize for the mistake in the submitted version of the manuscript and corrected the average and standard deviation values of WinsTC given in Table 1 (455 and 207 ng $g^{-1}$ instead of 459 and 204 ng $g^{-1}$). We also updated the WinsTC concentration of 2016 in Figure 3. The WinsTC deposition of 2016 and any conclusions made in the submitted version are not affected.

In Section 3.3.2, we noticed an error in the classification of the dust-laden samples to snow and rime. Previously, 7 snow and 1 rime sample were reported, while it should read 5 snow and 3 rime samples. We are sorry for the error in the submitted version of the manuscript and corrected it in the revised version. This change does not affect any conclusions made.

The legend in Figure 1 previously only showed "ATN" instead of "$ATN_{700-450}$". We clarified the parameter used here.

References

Baker, E. T. and Lavelle, J. W.: The effect of particle size on the light attenuation coefficient of natural suspensions, J. Geophys. Res.: Oceans, *89*, 8197-8203, https://doi.org/10.1029/JC089iC05p08197, 1984.

Kau, D., Greilinger, M., Kirchsteiger, B., Göndör, A., Herzig, C., Limbeck, A., Eitenberger, E., and Kasper-Giebl, A.: Thermal–optical analysis of quartz fiber filters loaded with snow samples–determination of

iron based on interferences caused by mineral dust, Atmos. Meas. Tech., 15, 5207-5217, https://doi.org/10.5194/amt-15-5207-2022, 2022.

Yamanoi, Y., Nakashima, S., and Katsura, M.: Temperature dependence of reflectance spectra and color values of hematite by in situ, high-temperature visible micro-spectroscopy, Am. Mineral., 94, 90–97, https://doi.org/10.2138/am.2009.2779, 2009.